# ^18^F-FDG PET/CT Findings in Cytomegalovirus Colitis

**DOI:** 10.3390/diagnostics9010003

**Published:** 2018-12-26

**Authors:** Anna Sophie L. Kjaer, Iben Ribberholt, Kim Thomsen, Per H. Ibsen, Elena Markova, Jesper Graff

**Affiliations:** 1Department of Infectious diseases, Copenhagen University Hospital Hvidovre, Kettegaards Allé 30, DK-2650 Hvidovre, Denmark; ibenribberholt@dadlnet.dk; 2Department of Clinical microbiology, Copenhagen University Hospital Rigshospitalet, Blegdamsvej 9, DK-2100 Copenhagen Ø, Denmark; kim.thomsen@regionh.dk; 3Department of Patology, Copenhagen University Hospital Hvidovre, Kettegaards Allé 30, DK-2650 Hvidovre, Denmark; per.ibsen@regionh.dk; 4Department of Radiology, Copenhagen University Hospital Hvidovre, Kettegaards Allé 30, DK-2650 Hvidovre, Denmark; elena.markova.01@regionh.dk; 5Department of Clinical Physiology & Nuclear Medicine, Copenhagen University Hospital Hvidovre, Kettegaards Allé 30, DK-2650 Hvidovre, Denmark; jesper.graff@regionh.dk

**Keywords:** ^18^F-FDG PET/CT, cytomegalovirus colitis, inflammatory bowel disease

## Abstract

We present a case demonstrating the diagnostic work-up of a patient undergoing azathioprine treatment for inflammatory bowel disease (IBD), diagnosed with an acute cytomegalovirus (CMV) infection and CMV colitis. An ^18^F-FDG positron emission tomography/computed tomography (PET/CT) performed 2 weeks after debut of symptoms revealed pathological ^18^F-FDG uptake in the left side of the colon mucosa, mimicked activity of IBD. However, a diagnosis of CMV colitis was based on the presence of CMV IgM antibodies, a seroconversion of CMV IgG antibodies, presence of CMV DNA in plasma and the finding af CMV DNA in biopsies from the intestinal mucosa. The patient responded to treatment with ganciclovir. This case highlights that a positive ^18^F-FDG PET/CT scan of the colon can be due to CMV colitis.

**Figure 1 diagnostics-09-00003-f001:**
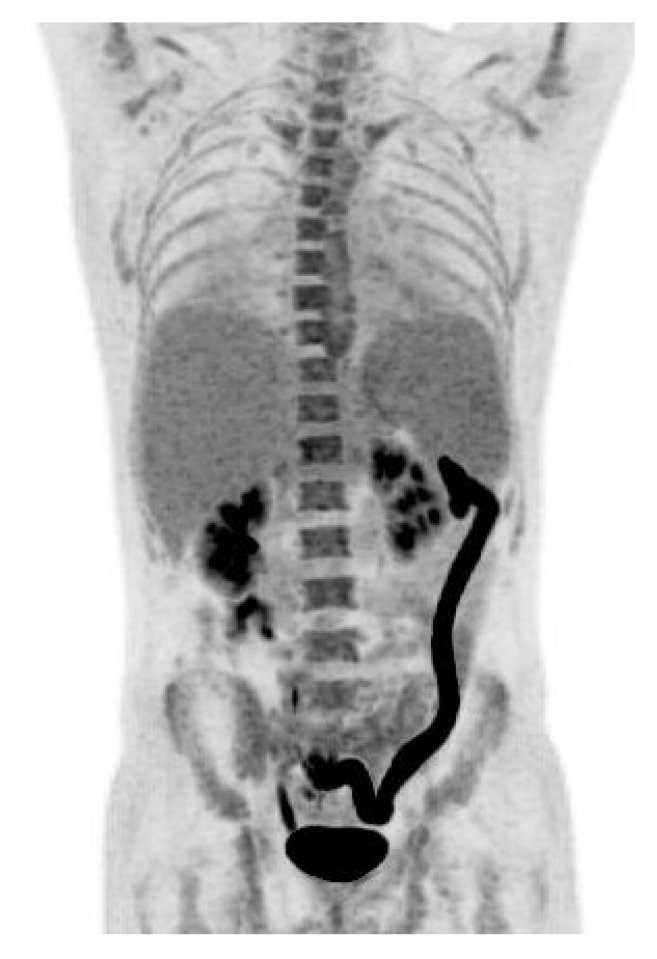
A 33 year-old man with known inflammatory bowel disease (IBD), on azathioprin (Imurel®) and mesalazin (Asacol®) treatment, presented with two weeks of hyperpyrexia up to 40.6 °C, dry cough, and water like diarrhoea which—unlike usual IBD—contained only a limited amount of blood. One month earlier, the patient had been travelling for 17 days in Jamaica. The patient was initially treated as an outpatient with roxithromycin for suspected pneumonia with no effect, and was admitted to the hospital after 10 days with symptoms. At admission, we found elevated C-Reactive Protein (CRP) 58 mg/L (normal range <10 mg/L), Alanine aminotransferase (ALAT) 217 U/L (normal range 10–70 U/L), Lactate dehydrogenase (LDH) 778 U/L (normal range 105–205 U/L), and Ferritin 2,560 µg/L (normal range 12–300 µg/L). Hemoglobin was low 6.5 mmol/L (normal range 8.3–9.5 mmol/L). White blood cell count was within normal range; 6.1 × 10^9^/L (normal range 3.5–8.8 ×10^9^/L). Blood cultures were negative. Human immunodeficiency (HIV) test was negative, hepatitis panels were either negative or consistent with previous vaccination. Dengue IgG and IgM, Zikavirus IgG and IgM and DNA, Chikungunya IgG and IgM and Malaria microscopy were all negative. A stool sample tested negative for bacteria, vira, and parasites. X-ray of the thorax was normal. CT of the abdomen revealed an enlarged descending colon with oedema of the colon mucosa and splenomegaly. The patient did not improve, and due to persistent pyrexia and watery diarrhoea, elevated ALAT and the finding of an enlarged spleen; Epstein Barr virus (EBV) and Cytomegalovirus (CMV) testing were ordered, especially as CMV reactivation have been found to be associated with IBD flare-up in patients on immunosupressive therapy. EBV serology was consistent with a previous infection; EBV DNA in the blood was negative. Specific CMV IgM antibodies were detected, and CMV IgG antibodies were borderline. However, only 3,000 copies/mL of CMV in the plasma were detected by PCR. At day 10 after admission, an ^18^F-FDG PET/CT scan was performed to identify the cause of fever of unknown origin (Figure 1). The scan revealed pathologically high ^18^F-FDG uptake in the left side of the colon, and mild diffuse increased activity in the bone marrow and in the enlarged spleen (16.5 cm). The intense ^18^F-FDG uptake in the left side of the colon was consistent with either flare-up in IBD or an infectious colitis. CMV testing was repeated on day 8 after admission, and this time both CMV IgG and IgM antibodies were detected, indicating a CMV IgG seroconversion consistent with a primary CMV infection. A sigmoidoscopy revealed inflammation with oedema and submucosal bleeding from 45 to 70 cm from the anus, which is not characteristic for IBD. The biopsy specimen from the colonic lesions did not reveal cells with CMV inclusion bodies, but PCR of the biopsies from the intestinal mucosa were positive for CMV DNA with 12,000 copies/mL suggesting compartmentalization of CMV manifestation. The patient was treated with ganciclovir 5 mg/kg twice daily for 14 days with good clinical response. Following treatment, CMV DNA in the blood was undetectable. CMV colitis is a known, but relatively rare complication to CMV infection that is observed in severely immunocompromised patients including those who have HIV and IBD, or have received an organ transplantation, chemotherapy, or other immunosuppressives [1,2,3]. CMV colitis as a complication in patients with IBD has been associated with active disease, immunosuppressive medication, steroid treatment, and especially steroid refractory disease progression [4]. Diagnosing CMV colitis is based on clinical symptoms, biochemical findings, typical endoscopic findings, histological examination of biopsies from colon mucosa, and detection of CMV DNA in the blood and biopsies from affected colon mucosa [1,2,3,4,5,6]. Discrimnation between activity in IBD and an acute CMV infectious colitis is difficult, but of great importance in order to initiate antiviral therapy. IBD patients with colitis and systemic signs of inflammation, steroid-refractory disease, pyrexia, splenomegaly, and a lack of leukocytosis have a high pre-test probability for CMV colitis [6] as was the case in our patient. ^18^F-FDG PET/CT has been shown to have a potential for a noninvasive whole-body assessment of IBD, with FDG accumulating along the intestinal tract including assessment of disease extent, activity, and treatment response [7]. In the present case, PET/CT scan revealed pathologically high ^18^F-FDG uptake in the left side of the colon, and mild diffuse increased activity in the bone marrow and in the enlarged spleen. The initial blood samples and the sigmoideoscopy did not indicate activity in the patients IBD, and the patient was diagnosed with a primary CMV colitis, based on CMV seroconversion and CMV DNA in the blood and the biopsy from the colon. The changes revealed by the PET/CT were consistent with an acute infectious colitis. Only one case of ^18^F-FDG uptake in acute CMV colitis has been previously reported [8]. This case highlights that a positive ^18^F-FDG PET/CT scan of the colon can be due to CMV colitis.

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
