# Peer review of "18F-FDG PET/CT Findings in Cytomegalovirus Colitis"

_diagnostics, 2018, doi:10.3390/diagnostics9010003_

Reviewer 1 Report

The authors present a case report of a patient who has inflammation in the colon visible by PET scan and noted by symptoms. The patient was thought to have IBD but antibody panel revealed the presence of CMV colitis.

1)The authors make a well-intentioned effort to convey their point but end up causing much confusion, for instance, the background is at the very end. It would help them to divide the text into paragraphs based on background, methods and results.

2) The English language needs to be checked and revised, for e.g. one is 'on' a treatment regimen not 'in' a treatment regimen and diagnosis is based on the 'presence' of an antibody screen, not on the 'present' of an antibody screen.

3)The authors have confirmed by day 3 that the CMV antibody test has come back positive so a diagnosis is already in place. Therefore, it is not clear why they seem alarmed by the FDG-PET scan, leading them to speculate IBD. The symptoms and case history indicate infection so why after 2 weeks do they think the patient has IBD. FDG is a well-known indicator of inflammation, but the cause of inflammation was clear from the serology panel. The patient also did not have a prior history of IBD and had just traveled to Jamaica and no point in the initial diagnosis did the authors suspect IBD. So why suspect IBD when the PET scan turns up positive after a CMV infection has already been confirmed. The rationale does not make sense.

Author Response

1)The authors make a well-intentioned effort to convey their point but end up causing much confusion, for instance, the background is at the very end. It would help them to divide the text into paragraphs based on background, methods and results.

The manuscript was submitted under the type of publication “Interesting Images”. We have followed the authors instruction as stated here:

No regular manuscript text (introduction/methods/results/discussion) should be included. Instead, images should be accompanied by detailed legends with no restriction in length. Reference citations should appear in the legends. Also, an unstructured abstract of no more than 200 words should be included as well as list of 3 to 10 keywords.

We therefore think that the format of the manuscript is in accordance with the journals instructions. However, we have rearranged the text to follow a more logical flow.

2) The English language needs to be checked and revised, for e.g. one is 'on' a treatment regimen not 'in' a treatment regimen and diagnosis is based on the 'presence' of an antibody screen, not on the 'present' of an antibody screen

We have now corrected the suggestions on word and spelling and a native speaking person have read the manuscript for English editing.

3)The authors have confirmed by day 3 that the CMV antibody test has come back positive so a diagnosis is already in place. Therefore, it is not clear why they seem alarmed by the FDG-PET scan, leading them to speculate IBD. The symptoms and case history indicate infection so why after 2 weeks do they think the patient has IBD. FDG is a well-known indicator of inflammation, but the cause of inflammation was clear from the serology panel. The patient also did not have a prior history of IBD and had just travelled to Jamaica and no point in the initial diagnosis did the authors suspect IBD. So why suspect IBD when the PET scan turns up positive after a CMV infection has already been confirmed. The rationale does not make sense.

The patient described in this case as stated in line 2 is known with IBD and in treatment with immunosuppressive therapy. This was the reason for suspecting IBD flare-up as reason for the symptoms and findings on PET/CT. The patient was diagnosed with a primary CMV infection based on specific CMV antibody seroconversion, positive CMV PCR in blood and tissue biopsies. Furthermore, the patient responded to anti-viral therapy.

We have now tried to emphases this findings by rewriting paragraphs in the manuscript.

We believe this has further improved the quality of the manuscript and hope it is now acceptable for publication.

Reviewer 2 Report

The authors present an example diagnosis of cytomegalovirus colitis using 18F-FDG PET/CT. Combined with immuno-diagnosis and circular blood DNA fragment analysis, the presence of CMV IgM and a increase level of IgG antibodies and DNA confirmed the cause of IBD is infection of CMV. The image is valuable for future diagnosis and can be served as a reference in such a rare cause of IBD. I support its publication in this journal.

Please correct line 23, diagnose should be diagnosis. Line 24, af should be of?

The manuscript needs editorial changes on wording and spelling.

Author Response

Thank you for your thorough editorial work on our case report.

We have now corrected the suggestions on word and spelling and a native speaking person have read the manuscript for English editing.

We believe this has improved the quality of the case report and hope it is now acceptable for publication.

Round  2

Reviewer 1 Report

The authors have made appropriate and necessary changes to manuscript. The new text added to the manuscript has improved it substantially. I thank the authors for an annotated copy for review and I trust that a clean copy (without red text) has been submitted to the journal editor for publication as my recommendation is to accept in present form.